# Recovery of Rare Earth Element from Acid Mine Drainage Using Organo-Phosphorus Extractants and Ionic Liquids

**Tommee Larochelle** [1,2], **Aaron Noble** [2,*], **Kris Strickland** [2], **Allie Ahn** [2], **Paul Ziemkiewicz** [3], **James Constant** [3], **David Hoffman** [3] **and Caitlin Glascock** [3]

1   L3 Process Development, Trois-Rivières, QC G9A 4M4, Canada
2   Virginia Tech, Blacksburg, VA 24061, USA
3   Water Research Institute, West Virginia University, Morgantown, WV 26506, USA
*   Correspondence: aaron.noble@vt.edu

**Abstract:** Acid mine drainage is a legacy environmental issue and one of the largest pollutants in many mining districts throughout the world. In prior work, the authors have developed a process for the recovery of critical materials, including the rare earth elements, from acid mine drainage using a preconcentration step followed by solvent extraction as a concentration and purification technology. As part of the downstream technology development efforts, we have synthesized a suite of ionic liquid extractants that facilitate greater separation factors leading to lower capital costs and reduced environmental impacts. This article provides a comparison of the conventional extractants D2EHPA, EHEHPA and C572 with their respective ionic liquids [c101][D2EHP,c101][EHEHP] and [c101][C572] for the recovery of rare earth elements from acid mine drainage. In the study, laboratory-scale, multi-contact solvent extraction tests were conducted at high and low extractant/dosages. The results show that the ionic liquids varied in performance, with [c101][D2EHP] and [c101][EHEHP] performing poorer than their conventional counterparts and [c101][c572] performing better. Recommendations for further study on [c101][c572] include stripping tests, continuous pilot testing, and techno-economic analysis.

**Keywords:** rare earth elements; solvent extraction; ionic liquids

## 1. Introduction

The transition of society toward more efficient and advanced technologies have revealed significant risk of disruption in the minerals supply chains. Some minerals, having specific technological applications with no or little replacement options and at risk of supply chain disruption have been termed "critical minerals" by various government and private entities [1–5]. While the methodologies for establishing criticality vary from country to country and agency to agency, all definitions encompass the notions of technological importance and supply chain disruption risk. One subset of elements containing many critical minerals is the lanthanide group, also referred to as rare earth elements.

The Rare Earth Elements (REEs) consist of 17 metals, specifically the 15 lanthanide elements, yttrium and scandium, which are typically found in the same mineral deposits as the lanthanides. The term "rare earths" originates from an old term used to designate minerals (earths) and their believed scarcity upon their initial discovery [6]. The rare earths designation is now considered a misnomer because it was demonstrated that the crustal abundance of some rare earth elements, such as cerium, is known to be higher than many common and widely used metals, such as copper [7].

Rare earth elements have chemical, physical, electrical, magnetic, optical, metallurgical, phosphorescent, nuclear, and catalytic properties that are often unique to them. The main properties linked to commercial applications of the REE are [8,9]:

- Strong magnetism,
- Fluorescent properties,
- Optical properties,
- Catalytic properties, and
- Glass polishing properties.

In an effort to diversity the supply of RE-containing materials, research efforts have recently focused on recovery of REEs from unconventional sources such as mine tailings [10–12], bauxite residues [13–15], electronic waste [16,17], and acid mine drainage [18–20]. Acid mine drainage has been shown to be an extremely promising feedstock given it's high content of magnet critical REEs as well as the net environmental benefits associated with its remediation [21,22].

The weathering of sulfide-bearing rock units in legacy mines often results in oxidation of the sulfides and generation of sulfuric acid [10]. The resulting acid reacts with a plethora of minerals, solubilizing them in the process [23]. Metal-laden discharge waters from these mines, referred to as acid mine drainage (AMD) have been demonstrated to contain a significant amount of rare earth elements [21,22]. Treatment of AMD is necessary prior to discharge and generally involves acid neutralization, oxidation and metal precipitation. Iron, aluminum, and manganese are the most common regulated metals in AMD. In prior work, the authors of this paper have developed and patented a method for the recovery of critical materials, including the rare earth elements, from AMD by modifying the treatment process to include a preconcentration step followed by solvent extraction leading to a high-grade product [24].

This process sequentially neutralizes AMD in a two-step sequence. First, the AMD is neutralized to a pH of 4.5 to precipitate nearly all iron and most of the aluminum. The resulting solution is further neutralized to a pH 8 to 8.5 to precipitate all REE with most cobalt and manganese as a pre-concentrate material. The REE/CM depleted water is sent to the permitted discharge point, meeting applicable clean water standards as set by the relevant local, state, and federal agencies. The authors have recently shown how the pre-concentrated material can be further refined to produce individually separated oxides and metals of high purity [25].

One approach being investigated for the recovery of REE from AMD involves concentrating the REE by staged precipitation [24] followed by sulfuric acid leaching and solvent extraction utilizing di(2-ethylhexyl) phosphoric acid (D2EHPA) in conjunction with tributyl phosphate (TBP) diluted in kerosene in a mixer-settler battery. The project team successfully operated this extraction circuit for over a year during the construction of a demonstration unit. However, the team noticed that D2EHPA while selective for REEs would also extract significant amounts of calcium and zinc [26].

A novel class of extractants: acid-base coupling task specific functional ionic liquids (ABC-TSFILs) derived from the neutralization of conventional acidic extractants using a basic cation extractant is being heavily investigated as an alternative to conventional acidic extractants. Ionic liquid extractants are attractive because their extraction mechanism does not rely on hydronium ion exchange. Because the extraction equilibrium of acidic REE extractants is a function of the hydronium ion concentration in the aqueous phase, it is often necessary to neutralize the pregnant leach solution between extraction stages to increase extraction efficiency [27]. In addition, the stripping of the organic to recover the rare earths requires high acidity, which requires neutralization during the precipitation process [27].

Another important benefit to the usage of ionic liquid extractants is a reduction of extractant losses in the raffinate. In his investigation, Su et al. [28] discovered that the solubility of [N1888][EHEHP] was approximately five times lower than that of EHEHPA in the same experiment.

Most published research on the extraction of rare earth elements using the Trihexyl (tetradecyl)phosphonium cation focuses on the ionic liquid extractant Trihexyl(tetradecyl) phosphonium Bis(2,4,4-trimethylpentyl)phosphinate [P66614][ BTMPP], also marketed as

Cyphos 104 IL. Similar to Cyanex 272 from which it is derived, Cyphos 104IL does not appear to be an effective extractant for REEs [29–32].

Much fewer publications are available with regard to the other combinations of acidic extractants and Trihexyl(tetradecyl)phosphonium. Sun et al. [33] investigated many di(2-ethylhexyl)phosphate [D2EHP] ionic liquids for the extraction of REEs in nitric acid media, in studies on the trivalent actinides/lanthanides separations by phosphorous-reagent extraction from aqueous complexes (TALSPEAK) process. Unfortunately the TALSPEAK process is not applicable to most industrial processes owing to its usage of nitric acid, glycolic acid and diethylenetriamine-N,N,N′,N″,N″-pentaacetic acid.

In other studies, Trihexyl(tetradecyl) phosphonium mono-(2-ethylhexyl) 2-ethylhexyl phosphonate was investigated alone and in conjunction with trioctylmethylammonium bis(2,4,4-trimethylpentyl) phosphonate [N1888][BTMPP] by Zhao et al. [34]. An unexpected synergistic effect was observed for the extraction of lutetium from chloride solutions.

Given the significant potential of ionic liquids and the lack of literature on the matter, we have investigated the extraction behavior of the conventional extractants Cyanex 801 and Cyanex 572 and compared them to our conventional D2EHPA-TBP extraction system for the recovery of REE from sulfuric acid AMD-derived pregnant leach solutions. We then prepared Trihexyl(tetradecyl) phosphonium ionic liquids with each of the three extractants and benchmarked the ionic liquids against the conventional extractant and among each other.

## 2. Materials and Methods

### 2.1. Material and Reagents

Di-(2-ethylhexyl) phosphoric acid (D2EHPA) and kerosene were purchased from Fisher Scientific, Waltham, MA, USA. Trihexyl(tetradecyl)phosphonium chloride (Cyphos IL 101) was obtained from Strem Chemicals, Inc, Newburyport, MA, USA. Tributyl phosphate (TBP) was purchased from MilliporeSigma. EHEHPA was supplied by Future Chemical Industry Co., Ltd., Shanghai, China. Cyanex 572 was obtained from Cytec Industries Inc., Toronto Canada. All chemicals were used as received without further purification.

D2EHPA, Cyanex 572, Cyphos 101, and EHEHPA were purchased to synthesize the ILs selected for this study. Trihexyl(tetradecyl)phosphonium chloride (Cyphos IL 101) is a high-purity phosphonium-based ionic liquid. Di-(2-ethylhexyl) phosphoric acid (D2EHPA) and 2-ethylhexyl phosphonic acid mono-2-ethylhexyl ester (EHEHPA) are acidic organophosphorus cation exchange extractants. D2EHPA was mixed with tributyl phosphate (TBP), which acts to improve extraction performance. Cyanex 572 is a highly stable mixture of phosphonic and phosphinic acids and acts as a chelating extractant. Kerosene was utilized as a diluent.

### 2.2. Ionic Liquids Synthesis

Three phosphonium ionic liquids were prepared: (1) [c101][c572], (2) [c101][EHEHP], and (3) [c101][D2EHP]. Each IL was synthesized at a concentration of 0.1 mol/L and 1 mol/L respectively. Kerosene was used as a diluent to reduce viscosity of the extractants. The phosphonium ionic liquids were prepared following a bicarbonate neutralization process described elsewhere [35] and used without further purification of separation.

### 2.3. Preparation of AMD-Derived Pregnant Leach Solution

The REE pregnant leach solution (PLS) was prepared by acid leaching of well mixed pre-concentrate material with a solids concentration of 1.5% solids. For this study, the pre-concentrate samples was produced from a typical Central Appalachian AMD source using the staged precipitation process described by Ziemkiewicz et al. [24]. Ten gallons of pre-concentrate were mixed with a Cole Parmer mixer at 80 RPM. While mixing, acid was added to the solution until the desired amount of dissolved TREE was achieved. The solution was then flocculated with a diluted polymer solution in the reaction vessel and mixed at 80 RPM for three minutes. The solution was then mixed at 30 RPM for an

additional 2 min before settled for 30 min. The supernatant was pumped out of the reaction vessel and through a 1-micron bag filter. The underflow slurry was filtered with the same bag filter to remove the leaching residuals from the solution.

The PLS filtrate was placed back in the reaction vessel. While mixing at 80 RPM, sodium hydroxide was added to the solution until 90% of the aluminum in the PLS solution precipitated out, which represents a target pH of 4.4. Once the solution was sufficiently neutralized to remove the aluminum, the flocculation method above was repeated. The supernatant was filtered through a new bag filter with a 1 micron pore size. This bag filter was also used to filter the slurry underflow. The neutralized PLS was then used for subsequent solvent extraction testing.

### 2.4. Extraction Procedure

Prior to solvent extraction, the initial pH of the PLS was measured and confirmed to be 4.4. This pH operating point is the natural pH of PLS derived from our AMD-derived preconcentrate leach and purification process. During the actual extraction tests, the aqueous phase solution was first poured into the organic phase and agitated using an impeller blade for 10 min or until the solution appeared homogenous and equilibrium was reached. After agitation, the solution was poured into a separatory funnel where the disengagement time and color of the phases were recorded. After separating the phases, the volume, pH, temperature, and oxidation-reduction potential were recorded for the raffinate solution. An aliquot of the raffinate was also retained for inductively coupled plasma mass spectrometry (ICP-MS) analysis to determine element concentrations (Agilent 7900). This process was repeated for four successive contacts, each using fresh leachate and the loaded organic resulting from the prior step. All stages were conducted at an organic to aqueous (O:A) ratio of 1:10. Raffinate samples from each contact were analyzed for elemental content, and the resulting data was then used to conduct a mass balance to determine the element concentrations of the organic phase solution. All experimental parameters were held constant throughout the four contacts.

### 2.5. Extraction Extent

The single stage extraction extent (%E) of an element or group of elements (*El*) is defined as the percentage of that element extracted to the organic phase on a mass basis. It is calculated using Equation (1).

$$Extraction\ Extent\ (\%E) = \frac{V_{PLS} * [El]_{PLS} - V_{Raf} * [El]_{Raf}}{V_{PLS} * [El]_{PLS}} \tag{1}$$

The cumulative extraction extent (*%Ec*) of an element or group of elements (*El*) is defined as the percentage of that element extracted to the organic phase on a mass basis considering all the extractions performed with the organic phase (n stages). It is calculated using Equation (2).

$$Cum.\ Extraction\ Extent,\ n\ stages\ (\%Ec) = \frac{n * V_{PLS} * [El]_{PLS} - \sum_{0}^{n} V_{Raf(i)} * [El]_{Raf(i)}}{n * V_{PLS} * [El]_{PLS}} \tag{2}$$

Given these equations and the experimental parameters provided above, negative extraction efficiencies are plausible and represent back extraction of elements from the loaded organic to the new PLS.

## 3. Results

### *3.1. Extraction Extent*

The single stage and cumulative extraction extent for the investigated extraction system are presented at Figures 1–6. In these figures SEG represents samarium, europium and gadolinium and HREE represents elements heavier than dysprosium such as holmium, erbium, ytterbium, thulium and lutetium.

The PLS composition is presented as Table 1.

**Table 1.** PLS Composition.

| Element | Concentration (mg/L) |
|---|---|
| Fe | <1.0 |
| Ca | 76.2 |
| Mg | 458.4 |
| Al | 41.2 |
| Zn | 260.0 |
| Th | <0.1 |
| U | 0.2 |
| Sc | 0.02 |
| Y | 39.9 |
| La | 8.7 |
| Ce | 22.3 |
| Pr | 4.5 |
| Nd | 22.2 |
| Sm | 6.4 |
| Eu | 1.6 |
| Gd | 9.7 |
| Tb | 1.4 |
| Dy | 7.7 |
| Ho | 1.4 |
| Er | 3.5 |
| Tm | 0.4 |
| Yb | 2.0 |
| Lu | 0.3 |

### 3.1.1. D2EHPA-TBP Extraction Data

The single stage and cumulative extraction extent for low concentration (0.1 mol/L) and high concentration (1.0 mol/L) D2EHPA-TBP extraction system are presented in Figure 1.

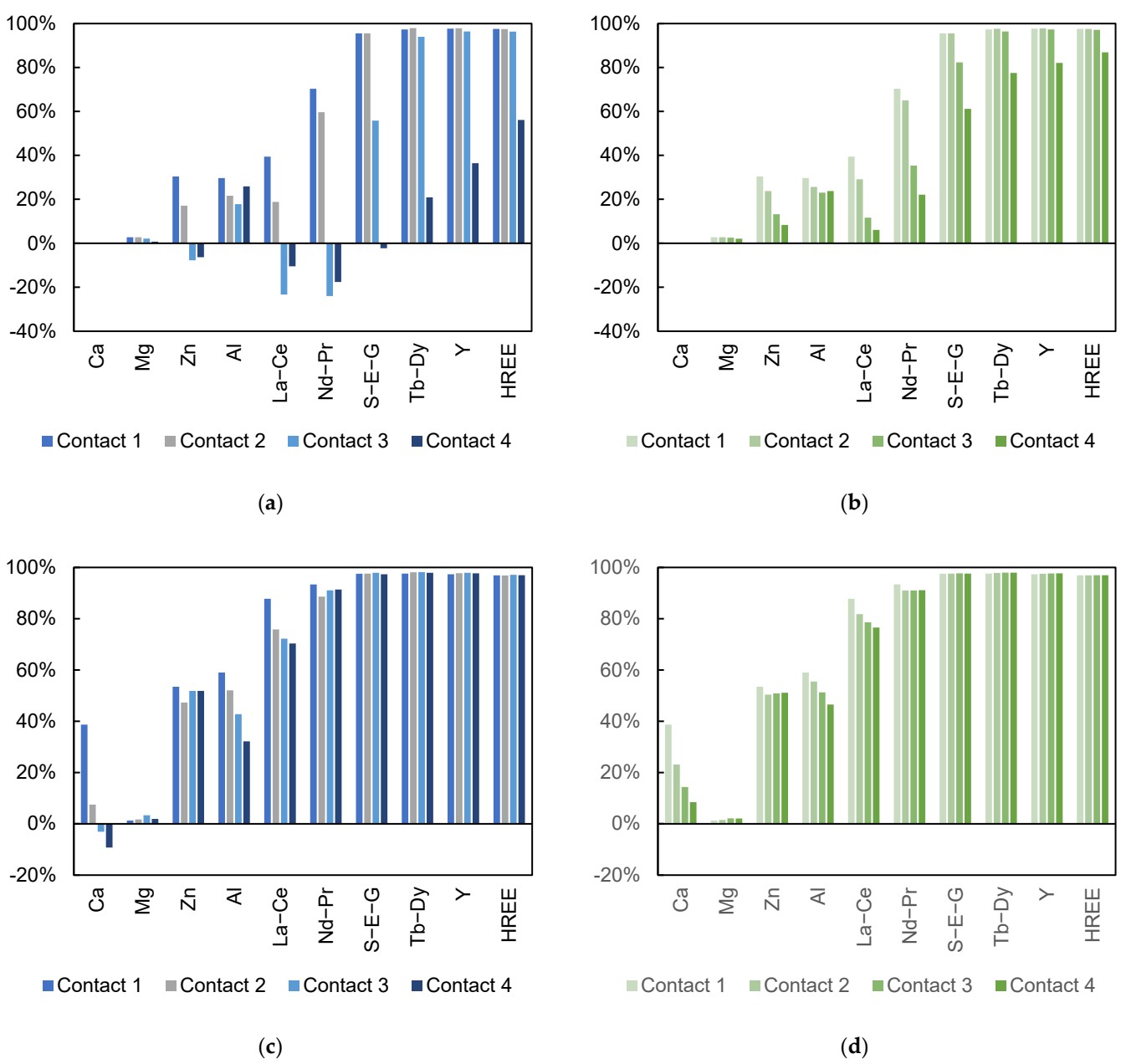

**Figure 1.** (**a**) 0.1 mol/L D2EHPA Single Stage Extraction Extent; (**b**) 0.1 mol/L D2EHPA Cumulative Extraction Extent; (**c**) 1 mol/L D2EHPA Single Stage Extraction Extent; (**d**) 1 mol/L D2EHPA Cumulative Extraction Extent.

### 3.1.2. EHEHPA Extraction Data

The single stage and cumulative extraction extent for low concentration (0.1 mol/L) and high concentration (0.9 mol/L) EHEHPA extraction system are presented as Figure 2.

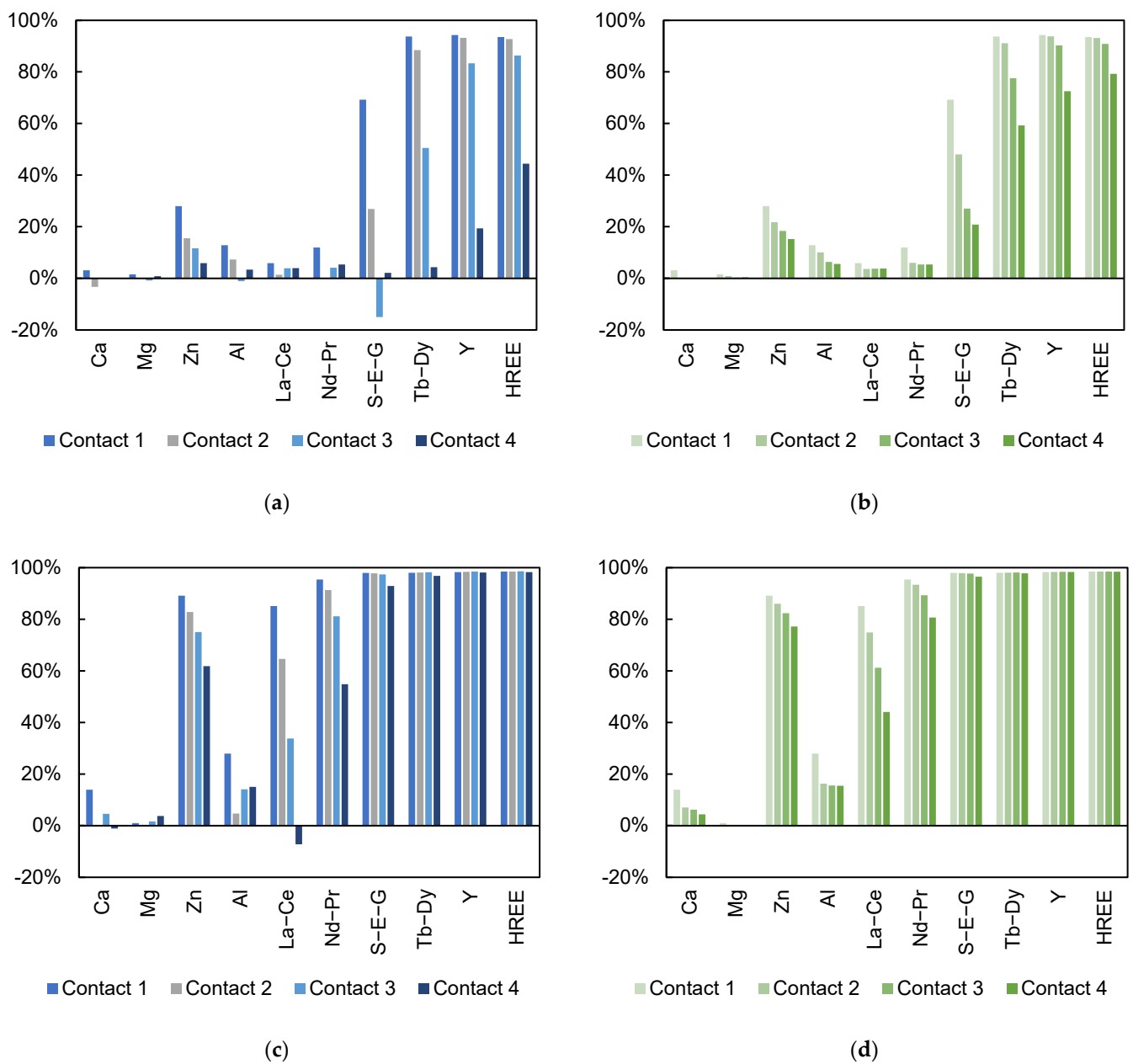

**Figure 2.** (**a**) 0.1 mol/L EHEHPA Single Stage Extraction Extent; (**b**) 0.1 mol/L EHEHPA Cumulative Extraction Extent; (**c**) 0.9 mol/L EHEHPA Single Stage Extraction Extent; (**d**) 0.9 mol/L EHEHPA Cumulative Extraction Extent.

### 3.1.3. C572 Extraction Data

The single stage and cumulative extraction extent for high concentration (1.05 mol/L) Cyanex 572 extraction system are presented as Figure 3.

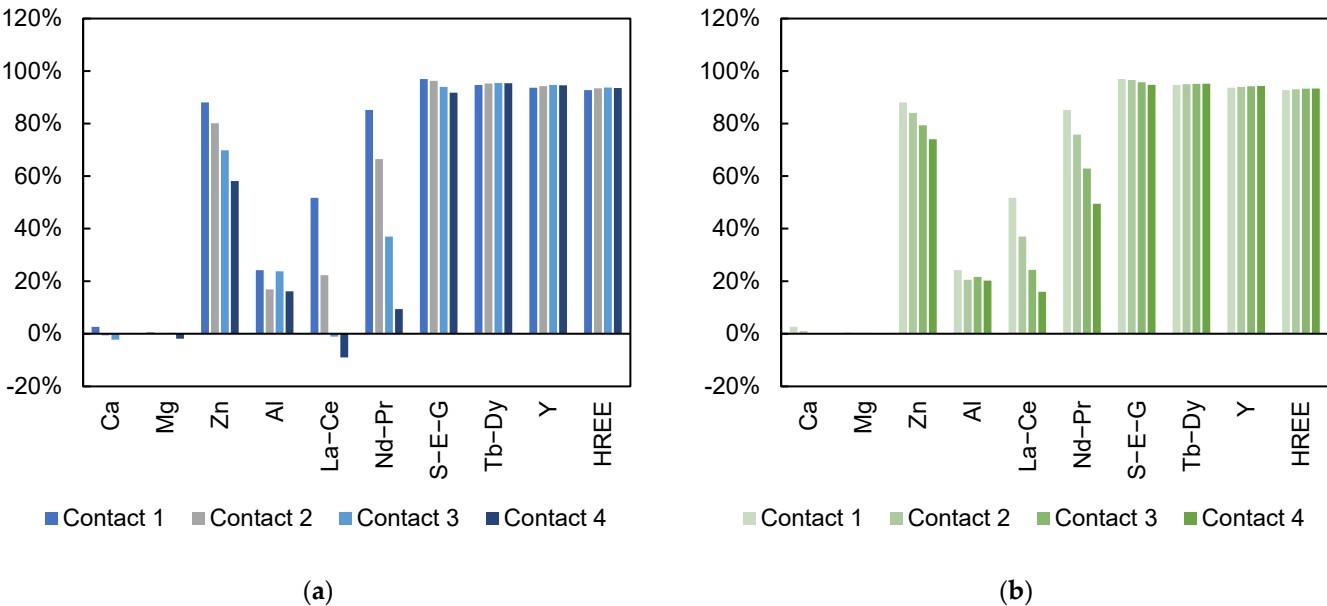

(**a**)                                             (**b**)

**Figure 3.** (**a**) 1.05 mol/L Cyanex 572 Single Stage Extraction Extent; (**b**) 1.05 mol/L Cyanex 572 Cumulative Extraction Extent.

### 3.1.4. [c101][EHEHP] Extraction Data

The single stage and cumulative extraction extent for low concentration (0.1 mol/L) and high concentration (1.0 mol/L) [c101][EHEHP] extraction system are presented as Figure 4.

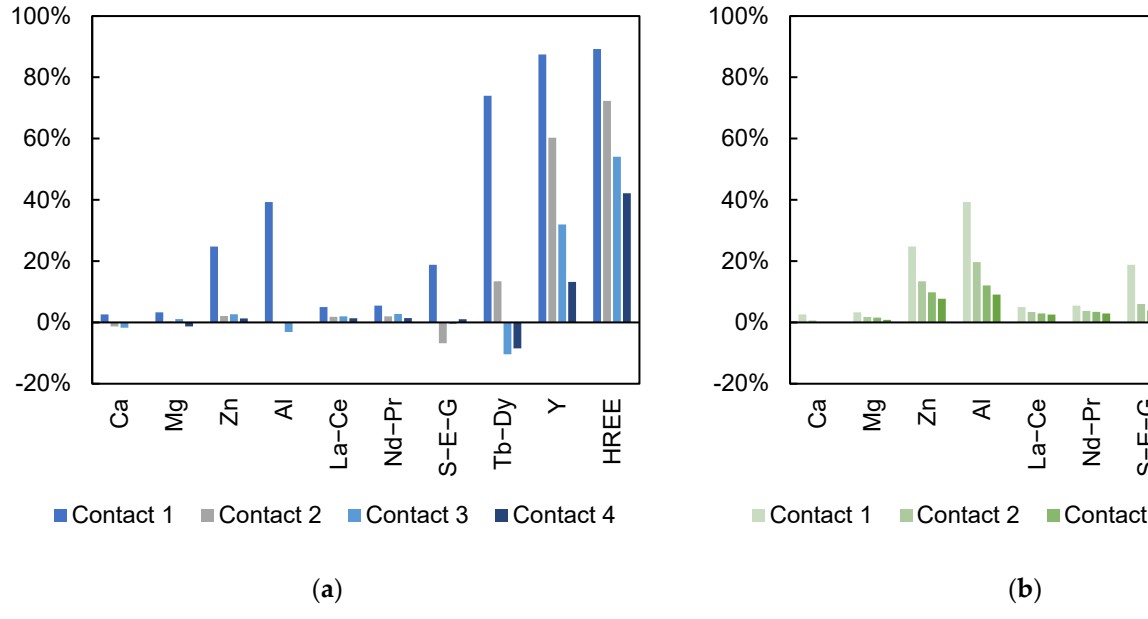

(**a**)                                             (**b**)

**Figure 4.** *Cont*.

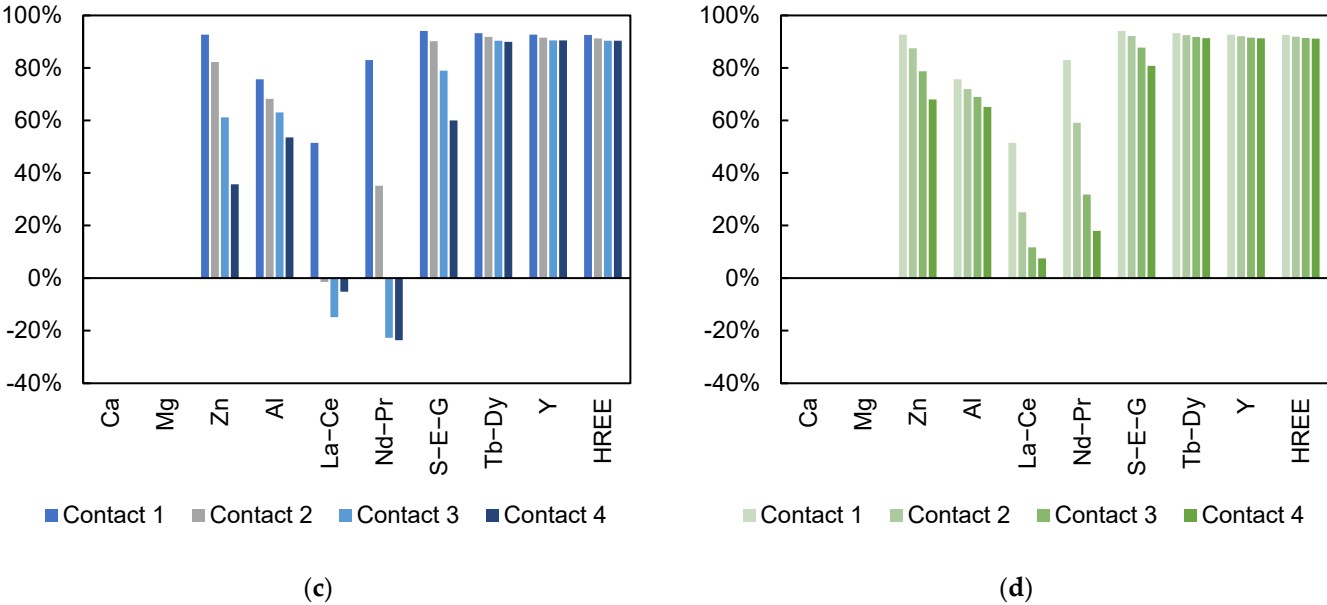

(**c**)                                                        (**d**)

**Figure 4.** (**a**) 0.1 mol/L [c101][EHEHP] Single Stage Extraction Extent; (**b**) 0.1 mol/L [c101][EHEHP] Cumulative Extraction Extent; (**c**) 1 mol/L [c101][EHEHP] Single Stage Extraction Extent; (**d**) 1 mol/L [c101][EHEHP] Cumulative Extraction Extent.

### 3.1.5. [c101][D2EHP] Extraction Data

The single stage and cumulative extraction extent for low concentration (0.1 mol/L) and high concentration (1.0 mol/L) [c101][D2EHP] extraction system are presented as Figure 5.

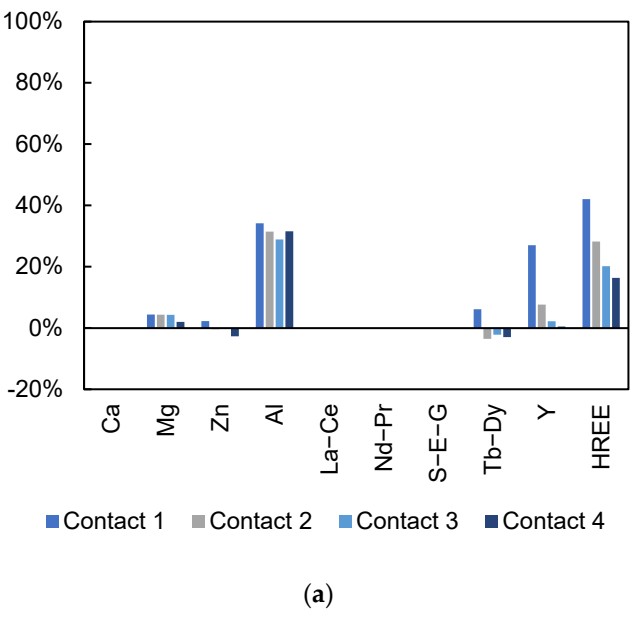

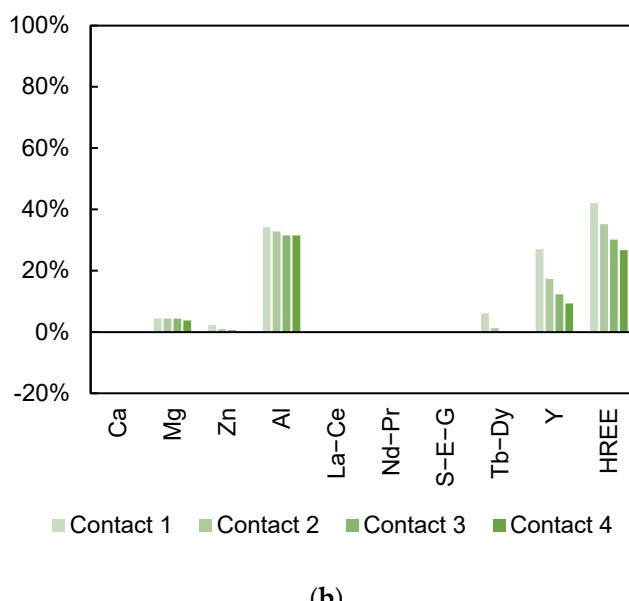

(**a**)                                                        (**b**)

**Figure 5.** *Cont.*

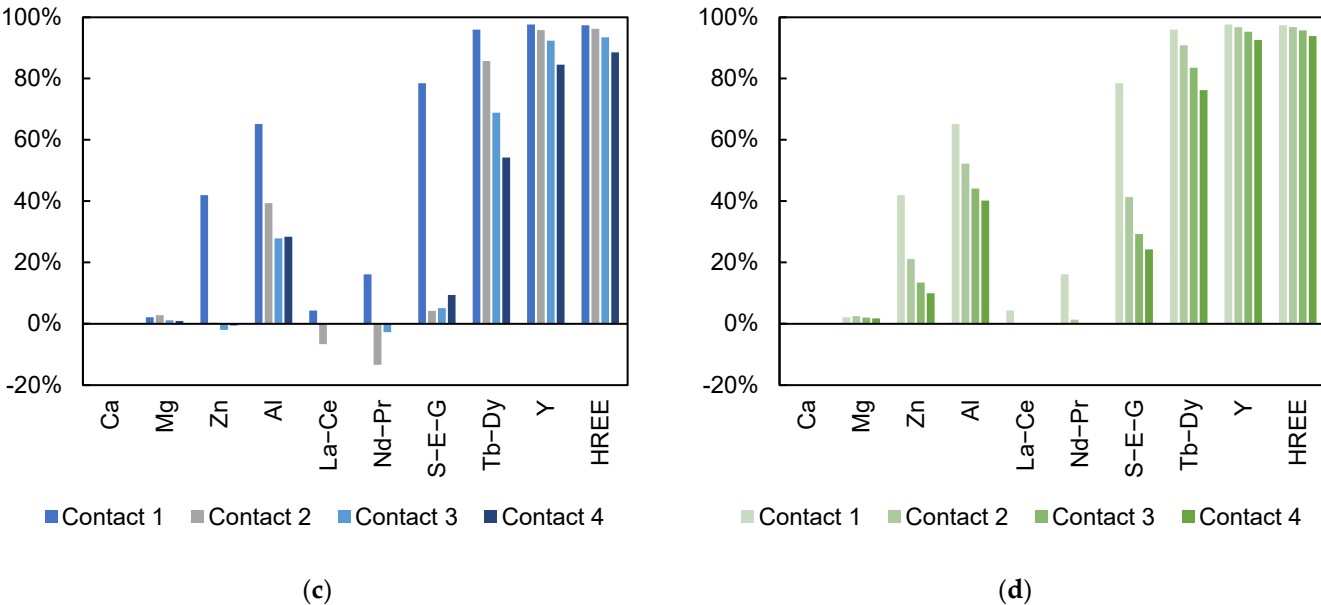

(**c**)　　　　　　　　　　　　　　　　　　(**d**)

**Figure 5.** (**a**) 0.1 mol/L [c101][D2EHP] Single Stage Extraction Extent; (**b**) 0.1 mol/L [c101][ D2EHP] Cumulative Extraction Extent; (**c**) 1 mol/L [c101][ D2EHP] Single Stage Extraction Extent; (**d**) 1 mol/L [c101][D2EHP] Cumulative Extraction Extent.

### 3.1.6. [c101][c572] Extraction Data

The single stage and cumulative extraction extent for low concentration (0.1 mol/L) and high concentration (1.0 mol/L) [c101][c572] extraction system are presented as Figure 6.

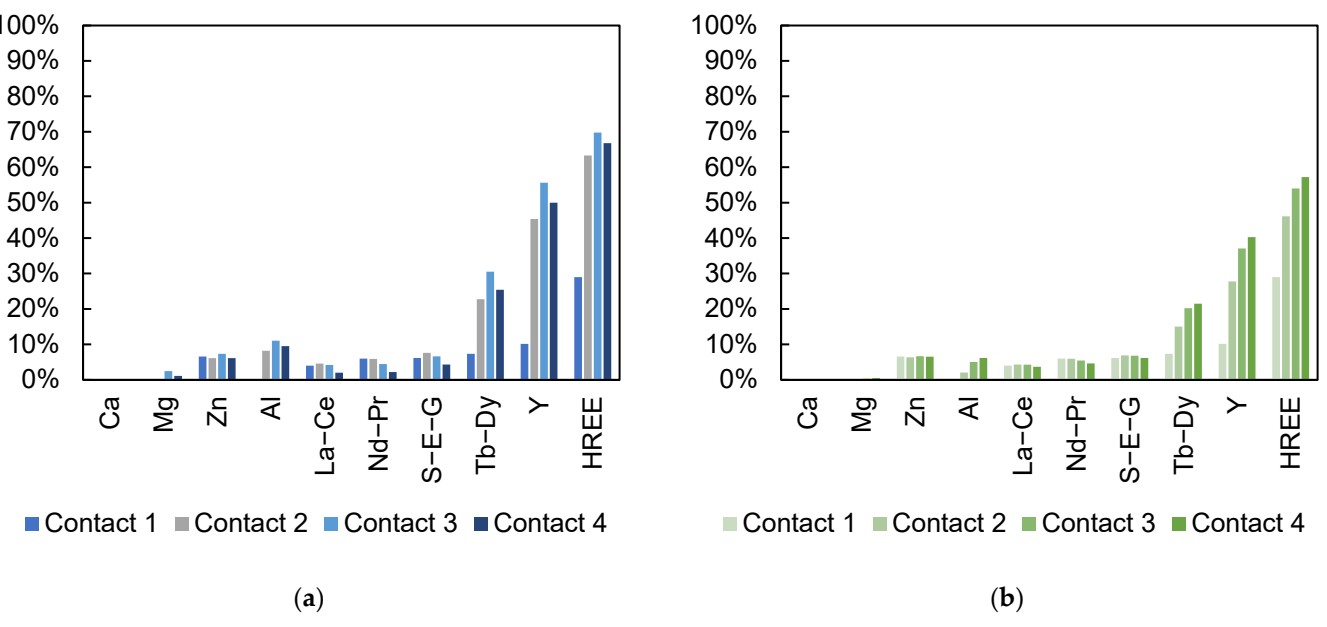

(**a**)　　　　　　　　　　　　　　　　　　(**b**)

**Figure 6.** *Cont*.

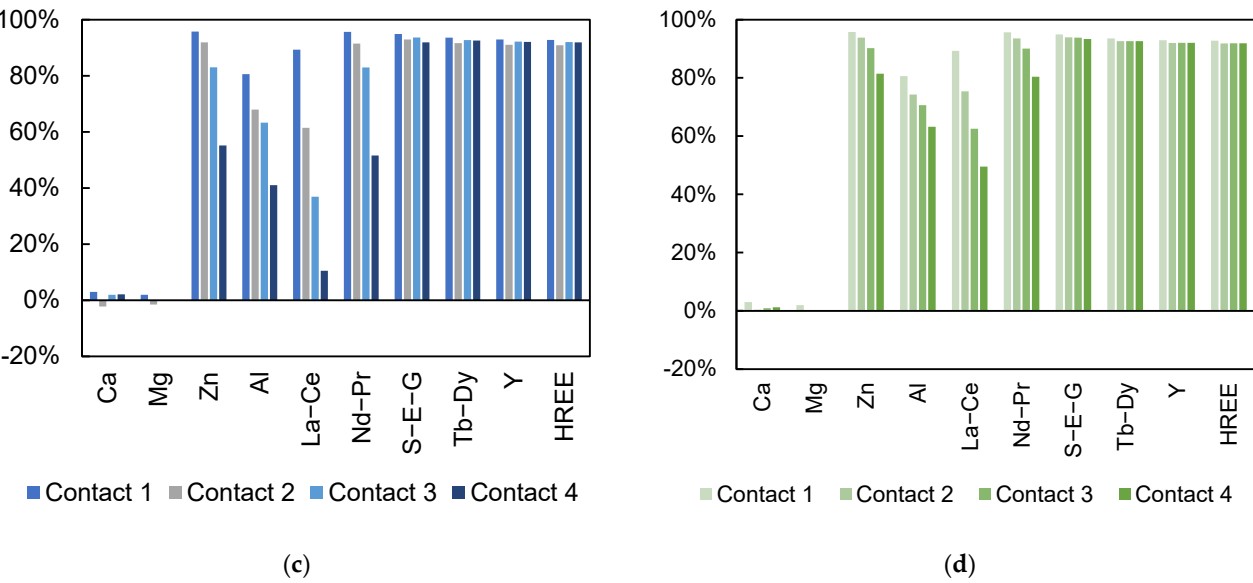

(**c**)                                                                                           (**d**)

**Figure 6.** (**a**) 0.1 mol/L [c101][c572] Single Stage Extraction Extent; (**b**) 0.1 mol/L [c101][c572] Cumulative Extraction Extent; (**c**) 1 mol/L [c101][c572] Single Stage Extraction Extent; (**d**) 1 mol/L [c101][c572] Cumulative Extraction Extent.

*3.2. Distribution Ratio*

The distribution ratio of an element is a measure of its affinity for one phase over another. It is defined as the concentration of the element in the organic phase divided by the concentration of that element in the aqueous phase as presented in Equation (3).

$$Distribution\ Ratio\ (D_{El}) = \frac{[El]_{Org}}{[El]_{Aq}} \tag{3}$$

The distribution ratio for a specific element (El) is a function of the elemental concentration of the system as well as of the composition of the organic phase. Distribution ratios of systems with excess extractant with regard to the concentration of elements to be extracted provide a more useful parameter for comparison between extraction systems. Distribution ratios for key elements of the investigated systems fourth contact in order to allow the distribution profile to develop are presented as Figure 7.

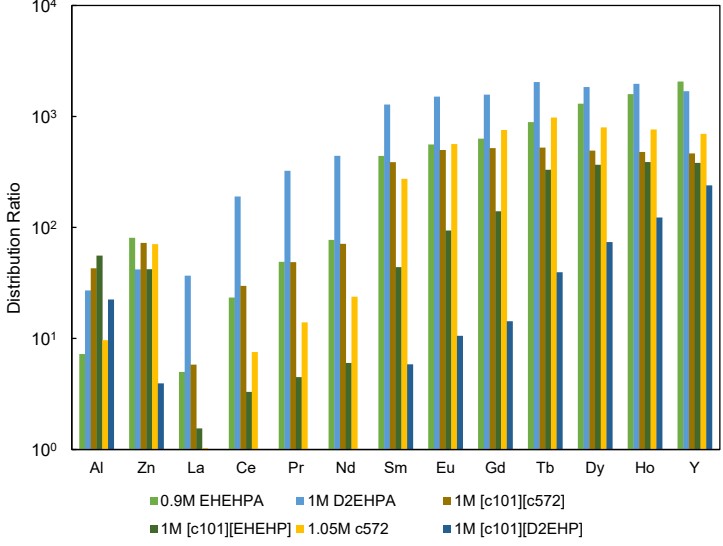

**Figure 7.** Concentrated Extraction System Distribution Ratios.

The data suggests that distribution ratio profiles were not fully established after the fourth contact for elements heavier than samarium in most extraction systems and thus cannot be reliably measured within our investigation. Additionally, the presence of a large concentration of zinc and aluminum likely also contributes to this effect. As such, separation factors usually derived from these distribution ratios were calculated from dilute extractant systems.

In order to understand the behavior of a system, it is crucial to understand the interdependencies between the various elements of that system. One such measure, the ratio of distribution ratios between two elements in a system, is termed "Separation Factor" and provides an indication of the relative affinity for the organic phase over the aqueous phase. It is defined as a ratio of distribution ratios, and is calculated using Equation (4). Separation factors are presented as Table 2.

$$Separation\ Factor\ (SF_{El1/El2}) = \frac{D_{El1}}{D_{El2}} \tag{4}$$

**Table 2.** Separation Factors.

| Element Pair | 0.1 mol/L EHEHPA | 0.1 mol/L [c101][EHEHP] | 0.1 mol/L D2EHPA | 0.1 mol/L [c101][c572] |
|---|---|---|---|---|
| Al/Pr | 1.38 | 4.26 | 2.24 | 1.81 |
| Zn/Pr | 3.89 | 3.67 | 0.55 | 1.81 |
| Ce/La | 2.04 | 1.10 | | 1.28 |
| Pr/Ce | 0.91 | 0.81 | 1.90 | 0.93 |
| Nd/Pr | 1.43 | 1.47 | 1.38 | 1.31 |
| Sm/Nd | 2.25 | 0.80 | 2.63 | 1.14 |
| Eu/Sm | 1.49 | 1.03 | 1.18 | 1.00 |
| Gd/Eu | 1.32 | 1.44 | 1.07 | 1.27 |
| Tb/Gd | 1.89 | 2.29 | 1.31 | 2.27 |
| Dy/Tb | 1.29 | 2.01 | 1.17 | 1.94 |
| Ho/Dy | 1.19 | 1.92 | 1.11 | 1.70 |
| Y/Ho | 1.17 | 1.70 | 1.15 | 1.50 |

The low extraction extents of 0.1 mol/L [c101][D2EHP] prevented the data from being used. Note that no tests were performed using a low concentration of Cyanex 572.

### 3.3. Organic Loading

Total metal loading in the organic represents a measure of the extraction capacity for an organic phase as a function of the aqueous phase composition. In traditional investigations, this data would be utilized to generate McCabe Thiele diagrams [36] and select a number of operating stages and a O:A ratio for the operation of the continuous circuit. In our investigation, we have compared the total organic loading of the various extractants to narrow down the scope of further investigations. The loading for key elements and the REEs after four PLS extractions is presented as Table 3. The total elemental loading for the extractants as a function of the contact number is presented as Figure 8.

**Table 3.** Elemental Loading (g/L).

| Element | TREE | Ca | Al | Zn | Total |
|---|---|---|---|---|---|
| D2EHPA | 4.64 | 0.21 | 1.06 | 5.31 | 9.99 |
| EHEHPA | 3.23 | 0.17 | 0.63 | 6.51 | 10.54 |
| C572 | 3.54 | 0 | 0.33 | 7.70 | 11.24 |
| [c101][D2EHP] | 2.11 | 0 | 0.91 | 1.03 | 3.14 |
| [c101][EHEHP] | 2.93 | 0 | 1.07 | 7.06 | 10.00 |
| [c101][c572] | 4.22 | 0.04 | 1.05 | 8.47 | 12.69 |

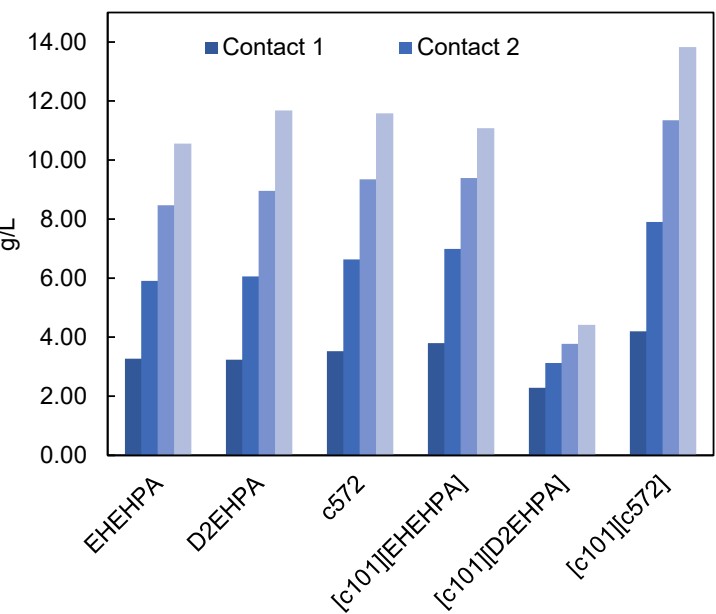

**Figure 8.** Total Organic Elemental Loading (in g/L).

The slope of all extractants in Figure 8 reveal that they are not fully loaded after 4 contacts and could extract more metal ions. However, [c101][c572] has more extraction capability than all other extractants, owing to its ability to extract more zinc than D2EHPA.

## 4. Discussion

The contrast between the low and high concentration of extractants provides insights on the applicability of the various extractants for the recovery of REE from AMD-derived PLS. The low concentration test allows comparison off the extraction efficiency between elements or groups of elements and reveals which are favored and if displacement can occur in the system. The degree of displacement is a measure of how well a battery of counter-current extraction stages will allow for the separation of various elements. It represents a crude approximation of the separation factors discussed in Section 3.3. As expected, all conventional acidic extractants performed well with very high extraction efficiencies, even with multiple contacts, and with a preference for HREEs. The data supports the pattern predicted by the acidic extractants acid dissociation constant (pka) in the extraction efficiency order of D2EHPA > EHEHPA > Cyanex 572 [37,38].

The key aspect of the experimental results is the large variation of affinity for calcium, aluminum and zinc observed between the acidic extractants. These three metals represent up to 80% of the metal loading in the organic and directly compete with the REEs. D2EHPA and EHEHPA are the only extractants which extract a significant amount of calcium from the PLS with approximately 40% and 15% extraction extent in the first contact respectively. The initial extraction and subsequent replacement of calcium by other metals is likely to result in the oversaturation of calcium sulfate followed by the precipitation of gypsum in the solvent extraction circuit. The presence of gypsum will likely reduce the phase separation efficiency in the settlers and could result in clogged piping, unbalancing the circuit hydraulics. Unfortunately, the operation of D2EHPA in a counter-current circuit will only exacerbate the gypsum precipitation problem. Our findings suggest that D2EHPA should not be used for this application and that Cyanex 572 should be favored since its affinity for calcium is markedly lower than EHEHPA without presenting a very different extraction efficiency for the other elements.

A review of the ionic liquid extraction efficiency reveals that both [c101][D2EHP] and [c101][EHEHP] performed poorly in this system with [c101][D2EHP] performing particularly poorly with regard to both its selectivity for REEs and its overall extraction efficiency. While [c101][EHEHP] offered better extraction efficiencies for REEs, it did

not perform as well as [c101][c572]. From all the ionic liquids investigated in this study, [c101][c572] is the only one that revealed a potential for replacing conventional acidic extractants in the bulk sulfate extraction circuit. [c101][c572] offered extraction efficiencies similar to that of Cyanex 572, with lower affinity for aluminum, and lanthanum than neodymium and praseodymium while having no extraction capability for calcium.

All extractants offered low separation factors, with D2EHPA having the highest separation factors for LREE and [c101][EHEHP] the highest separation factors for HREEs, followed closely by [c101][c572]. D2EHPA is primarily used in the industry to separate LREE [33], but would be an uneconomic choice in a system such as the one studied, owing to its high HREE concentration. The stripping of HREE from D2EHPA requires high acidity [39] and is not practiced commercially [27]. Interestingly, [c101][c572] offered separation factor significantly higher than reported in the c572 literature for chloride media. Cytec, the supplier of Cyanex 572 [40] reports separation factors of 1.7 for the pair Dy/Tb and of 1.3 for the pair Ho/Dy, much lower than determined experimentally in our investigation.

## 5. Conclusions

The high extraction efficiency, very high loading and high separation factors of [c101][c572], comparable to the strongest commercial extractant: D2EHPA suggest that [c101][c572] may be a very good candidate for the replacement of conventional acidic extractant in the bulk extraction of REEs from AMD-derived PLS in sulfate media. The present work revealed that both zinc and calcium are significant detrimental elements in the current process and should be the subject of specific investigations in order to minimize their presence in the REE extraction circuit. The performance of all extractants is likely to be improved in the absence of these impurities. The performance of [c101][c572] was 13% superior to Cyanex 572, 20% superior to EHEHPA and 27% superior to D2EHPA the current commercially used extractants. In addition, the separation factors of [c101][c572] were significantly higher than D2EHPA and EHEHPA with respect to HREE separation, a large fraction of the AMD-derived feedstock distribution. As such, further investigations should evaluate the effect of operating parameters such as pH, temperature, extractant concentration and O:A ratio to design a continuous extraction circuit using the McCabe Thiele method. Stripping should also be evaluated using different acids and acidity levels. Finally, a techno-economic analysis should be performed to evaluate the economic incentive of using ionic liquids in this system.

**Author Contributions:** Conceptualization, T.L.; methodology, T.L., K.S. and A.A.; validation, T.L. and A.N.; formal analysis, T.L.; investigation, T.L., K.S. and A.A.; resources, D.H., C.G. and J.C.; writing—original draft preparation, T.L., K.S. and A.A.; writing—review and editing, A.N. and P.Z.; visualization, K.S.; supervision, A.N. and P.Z.; project administration, A.N. and P.Z. All authors have read and agreed to the published version of the manuscript.

**Funding:** This article is based upon work supported by the United States Department of Energy under Award DE-FE0031834with support from Virginia Tech Open Access Subvention Fund (VT's OASF) in publishing this article.

**Data Availability Statement:** Due to confidentiality agreements, supporting data can only be made available to bona fide researchers subject to a non-disclosure agreement. Details of the data and how to request access are available from Aaron Noble, aaron.noble@vt.edu at Virginia Polytechnic and State University.

**Conflicts of Interest:** The authors declare no conflict of interest.

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
