# Peer review of "Recovery of Rare Earth Element from Acid Mine Drainage Using Organo-Phosphorus Extractants and Ionic Liquids"

_minerals, doi:10.3390/min12111337_

Round 1

Reviewer 1 Report

The present work aimed the comparison of traditional organic extractants and ionic liquids for separation of rare earth elements from acid mine drainage. The study contributes to the development of technologies to obtain rare earth elements from secondary sources. The topic is important but miss literature review and comparison to similar studies using organo-phosphorus extractants to separate rare earth elements. The methodology is well written, but discussion must be improved in light of the recent literature.

Abstract

It is not described how the experiments were carried out. More than a half of the paragraph is general information that may be hidden.

Introduction

This section is too long. The information must be shorted since there are few points missed.

Page 1, line 34-35: the term “critical material/metal” is well described by the European Union (https://ec.europa.eu/docsroom/documents/42881, https://ec.europa.eu/docsroom/documents/42849). In addition, several countries have list of critical or strategic materials depending on their needs, such as Brazil, USA, and Australia (https://www.ga.gov.au/scientific-topics/minerals/critical-minerals#heading-1, https://www.energy.gov/sites/prod/files/2021/01/f82/DOE%20Critical%20Minerals%20and%20Materials%20Strategy_0.pdf, https://doi.org/10.1016/j.jenvman.2021.113091). So, the term is well-described around the world, and the authors must rewrite the sentence.

The authors should present a paragraph about studies regarding the recovery of rare earth elements from secondary sources, such as bauxite residue (https://link.springer.com/10.1007/s40831-021-00434-3), nickel laterite (http://link.springer.com/10.1007/s11837-019-03427-6) and silica-based ores (https://www.mdpi.com/2075-4701/12/7/1133).

Page 2 line 64-65: is there any loss of rare earth elements as iron and aluminium precipitated? Despite the authors cited their publication, for the purpose of this manuscript is important to clarify this point.

Materials and methods

Page 4 line 172-173: why SX experiments should carried out at pH 4.4?

Page 4 line 183: “The” is spelled wrong at the end of the sentence.

Results

Equation 1 should be presented in Materials and Methods

Figure 1: how extraction percentage is negative? (see Fig 1a for La, Ce, Nd, and Fig 1c for Ca). The same was observed in Fig 2-5.

What is S-E-G? Is Sm, Eu and Gd elements?

What is the composition of the solution?

Discussion

It is missed the comparison with the literature for the use of the same organic extractants or similar (such as https://doi.org/10.1016/j.seppur.2021.120223, https://doi.org/10.1016/j.jece.2019.103528, https://doi.org/10.1016/j.seppur.2021.119798, https://doi.org/10.1016/j.hydromet.2020.105463)

The use of organo-phosphorus extractants for rare earth elements must be highlighted, since these compounds have high selectivity (http://dx.doi.org/10.1016/j.mineng.2013.10.021)

Page 13 line 294: there is a letter missed.

The results are not compared between the organic extractants. What occurred when its concentration increased?

Conclusions

What are the main conclusions of the present study? It is missed. No quantitative data is presented to point the authors’ final observations. Is there any recommendation for future studies?

Author Response

Thank you you time and effort reviewing our manuscript. Please see our responses to your very valuable and insightful review.

Abstract

It is not described how the experiments were carried out. More than a half of the paragraph is general information that may be hidden.
---------------
Clarification added.

Introduction

This section is too long. The information must be shorted since there are few points missed.
---------------------
This comment is valid, however in this case we feel it is commensurate with relevant liteature where a stronger contextualisation is required

Page 1, line 34-35: the term “critical material/metal” is well described by the European Union (https://ec.europa.eu/docsroom/documents/42881, https://ec.europa.eu/docsroom/documents/42849). In addition, several countries have list of critical or strategic materials depending on their needs, such as Brazil, USA, and Australia (https://www.ga.gov.au/scientific-topics/minerals/critical-minerals#heading-1, https://www.energy.gov/sites/prod/files/2021/01/f82/DOE%20Critical%20Minerals%20and%20Materials%20Strategy_0.pdf, https://doi.org/10.1016/j.jenvman.2021.113091). So, the term is well-described around the world, and the authors must rewrite the sentence.
------------
Reworded to account for the comment.

The authors should present a paragraph about studies regarding the recovery of rare earth elements from secondary sources, such as bauxite residue (https://link.springer.com/10.1007/s40831-021-00434-3), nickel laterite (http://link.springer.com/10.1007/s11837-019-03427-6) and silica-based ores (https://www.mdpi.com/2075-4701/12/7/1133).
------
The authors appreciate the suggestion for additional citations, as we believe this improved the introductory discussion in our manuscript.  We identified well-cited articles on the various secondary sources and included them in our literature review

Page 2 line 64-65: is there any loss of rare earth elements as iron and aluminium precipitated? Despite the authors cited their publication, for the purpose of this manuscript is important to clarify this point.
---------------
REE losses with the Fe and Al precipitate is an intriguing topic that has been recently studied by Li et al.  For the purposes of this paper, though, the authors feel that the upstream REE recovery is not a significant factor in the solvent extraction performance.  By analogy, most solvent extraction papers using conventional ore feedstocks do not discuss the upstream recovery in flotation or other beneficiation processes, but instead begin the scope of their study with the resultant leach solution.  We believe our paper should follow a similar pattern, starting the scope of the study with the pregnant leach solution.  To that end, we do agree that the leachate composition should be included, as you have suggested. (edited) 
https://www.sciencedirect.com/science/article/pii/S092777572200317X

Materials and methods

Page 4 line 172-173: why SX experiments should carried out at pH 4.4?
------------
Added explanation.

Page 4 line 183: “The” is spelled wrong at the end of the sentence.
-------------------
Corrected

Results

Equation 1 should be presented in Materials and Methods
----------
Moved

Figure 1: how extraction percentage is negative? (see Fig 1a for La, Ce, Nd, and Fig 1c for Ca). The same was observed in Fig 2-5.
------------
This explanation was provided and is now in line 203 of the revised narative. "Given these equations and the experimental parameters provided above, negative extraction efficiencies are plausible and represent back extraction of elements from the loaded organic to the new PLS."

What is S-E-G? Is Sm, Eu and Gd elements?
---------------------
Added description of SEG and HREE.

What is the composition of the solution?
---------------------
Added Table 1.

Discussion

It is missed the comparison with the literature for the use of the same organic extractants or similar (such as https://doi.org/10.1016/j.seppur.2021.120223, https://doi.org/10.1016/j.jece.2019.103528, https://doi.org/10.1016/j.seppur.2021.119798, https://doi.org/10.1016/j.hydromet.2020.105463)
---------------------
The authors agree that other extractants could be evaluated, however SX data is system specific and in this case there is no comparative litterature when real solution with similar quantities of Al and Zn are present, which is why we had to also test the acidic extractant to compare with ILs. The referenced papers do no use a similar PLS. 

The use of organo-phosphorus extractants for rare earth elements must be highlighted, since these compounds have high selectivity (http://dx.doi.org/10.1016/j.mineng.2013.10.021)
---------------------
Mention added to intro.

Page 13 line 294: there is a letter missed.
-------------------
Corrected

The results are not compared between the organic extractants. What occurred when its concentration increased?
-------------
The goal of the study was to determine which if any ionic liquid should be further investigated, this mechanism question will be addressed in a further study with synthetic material. We do not feel this can be properly addressed using real solutions containing large amounts of Zn.
A sentence was added to the conclusion.

Conclusions

What are the main conclusions of the present study? It is missed. No quantitative data is presented to point the authors’ final observations. Is there any recommendation for future studies?
--------------
Added

Reviewer 2 Report

This article shows a comparison study with different extractants and ionic liquids for the recovery of REE, performed on acid rock drainage. The study is very well structured and well written in general. The methodology is clearly presented and the analysis of results justifies the conclusions derived.

Some minor comments on aspects which must be corrected:

1-Line 155: "...(PLS) was prepared by acid leaching well mixed pre-concentrate (at 1.5% solids", please revise and complete

2-Line 168 "...filtered through a new 1-micron bag filter", please rephrase to indicate that the pore size is under 1 micron, if this is the meaning. Similar comment in Line 162.

3-Line 228: Figure 5?

4-Line 234: Figure 6?

5-Line 261: Scheme 1 is missing

6-Line 267: "McCabe Thiele diagrams", please insert proper reference

7-Please homogenize decimal symbol in the document (Table 1 and Table 2 show different criteria)

8-Line 294: "...between he acidic extractants". Please correct "he"

9- References: web reference for Cytec data sheet should be indicated, if available 

Author Response

Thank you for the comments.

1-Line 155: "...(PLS) was prepared by acid leaching well mixed pre-concentrate (at 1.5% solids", please revise and complete - Corrected

2-Line 168 "...filtered through a new 1-micron bag filter", please rephrase to indicate that the pore size is under 1 micron, if this is the meaning. Similar comment in Line 162.- Corrected

3-Line 228: Figure 5?- Corrected

4-Line 234: Figure 6?- Corrected

5-Line 261: Scheme 1 is missing- Corrected

6-Line 267: "McCabe Thiele diagrams", please insert proper reference - Added

7-Please homogenize decimal symbol in the document (Table 1 and Table 2 show different criteria) - We feel it would overload the table 2 to add 2 decimals. It is customary to report separation factors using 2 decimal for Table 1.

8-Line 294: "...between he acidic extractants". Please correct "he"- Corrected

9- References: web reference for Cytec data sheet should be indicated, if available, unfortunately not available.

Round 2

Reviewer 1 Report

The quality of the manuscript has improved and a few more revisions are required.

Page 1 line 35-40: the reference is missed.

Introduction is still too long.

I recommend the use of “mol/L” instead of “M”

My previous question about negative numbers were not answered, as Fig 5c (is the opposite reaction occurred?)

Author Response

Thanks for the additional comments

Page 1 line 35-40: the reference is missed.

---------

Do you have a specific location? I don't see any issue.

Introduction is still too long.

-----------

Paragraphs on Zinc and Calcium were removed.

I recommend the use of “mol/L” instead of “M”

-----------

Changes made

My previous question about negative numbers were not answered, as Fig 5c (is the opposite reaction occurred?)

------------

Line 188 to 190, yes negative extraction represent back extraction.
